# Adversarial Feature Learning under Accuracy Constraint for Domain Generalization

**Kei Akuzawa, Yusuke Iwasawa & Yutaka Matsuo**
School of Engineering
The University of Tokyo
{akuzawa-kei,iwasawa,matsuo}@weblab.t.u-tokyo.ac.jp

## Abstract

Learning domain-invariant representation is a dominant approach for domain generalization. However, previous methods based on domain invariance overlooked the underlying dependency of classes on domains, which is responsible for the trade-off between classification accuracy and the invariance. This study proposes a novel method *adversarial feature learning under accuracy constraint (AFLAC)*, which maximizes domain invariance within a range that does not interfere with accuracy. Empirical validations show that the performance of AFLAC is superior to that of baseline methods, supporting the importance of considering the dependency and the efficacy of the proposed method to overcome the problem.

## 1 Introduction

In supervised learning we typically assume that samples are obtained from the same distribution in training and testing; however, because this assumption does not hold in many practical situations it reduces the classification accuracy for the test data (Torralba & Efros, 2011). One typical situation is domain generalization (DG) (Blanchard et al., 2011; Shankar et al., 2018; Sriram et al., 2018; Erfani et al., 2016): we have labeled data from several source domains and collectively exploit them such that the trained system generalizes to other unseen, but somewhat similar, target domain(s).

This paper considers DG under the situation where domain $d$ and class $y$ labels are statistically dependent owing to some common latent factor $z$ (Figure 1-(c)), which we referred to as *domain-class dependency*. For example, the WISDM Activity Prediction dataset (WISDM, Kwapisz et al. (2011)), where $y$ and $d$ correspond to activities and wearable device users, exhibits this dependency because (1) some activities (e.g., jogging) are strenuous to the extent that some unathletic subjects avoided them (*data characteristics*), or (2) other activities were added only after the study began and the initial subjects could not perform them (*data-collection errors*). The dependency is common in real-world datasets (Zhang et al., 2013) and a similar setting has been investigated in domain adaptation (DA) studies, but most prior DG studies overlooked the dependency.

Most prior DG methods utilize invariant feature learning (IFL) (e.g., Muandet et al. (2013)). IFL attempts to learn feature representation $h$ from input data $x$ which is invariant to $d$. When source and target domains have some common structure (see, Muandet et al. (2013)), IFL prevents the classifier from overfitting to source domains (Figure 1-(b)). However, under the dependency, merely imposing the domain invariance can adversely affect the classification accuracy as pointed out by Xie et al. (2017) and illustrated in Figure 1. Although that trade-off occurs in source domains (because DG uses only source data during optimization), it can also negatively affect the classification performance for target domain(s). For example, if the target domain has characteristics similar (or same as an extreme case) to those of a certain source domain, giving priority to domain invariance obviously interferes with the DG performance (Figure 1-(d)).

In this paper, considering that prioritizing domain invariance under the trade-off can negatively affect the DG performance, we propose a novel method *adversarial feature learning under accuracy constraint (AFLAC)*, which maximizes domain invariance within a range that does not interfere with the classification accuracy (Figure 1-(e)) on adversarial training. Specifically, AFLAC is intended to achieve *accuracy-constrained domain invariance*, which we define as the maximum $H(d|h)$ ($H$ denotes entropy) value under the condition $H(y|x) = H(y|h)$ ($h$ has as much $y$ information as $x$).

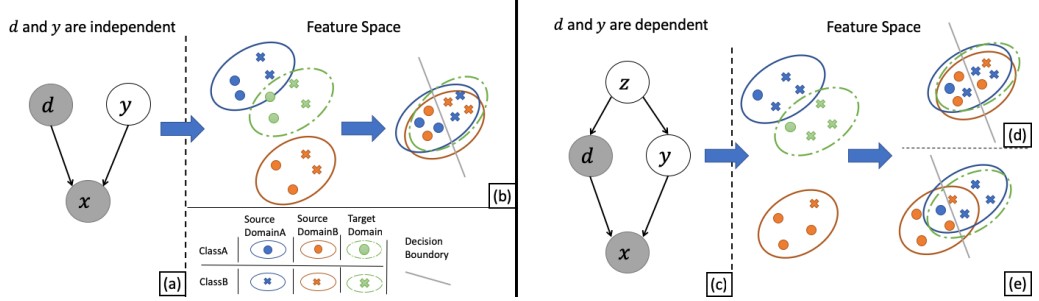

Figure 1: Explanation of domain-class dependency and the induced trade-off. (a) When the domain and the class are independent, (b) domain invariance and classification accuracy can be optimized at the same time, and the invariance prevents the classifier from overfitting to source domains. (c) When they are dependent, a trade-off exists between these two: (d) optimal classification accuracy cannot be achieved when perfect invariance is achieved, and (e) vice versa.

Empirical validations show that the performance of AFLAC is superior to that of baseline methods, supporting the importance of considering domain-class dependency and the efficacy of the proposed approach for overcoming the issue.

## 2    RELATED WORK

DG has been attracting considerable attention in recent years, and most prior DG methods utilize IFL (Muandet et al., 2013; Erfani et al., 2016; Ghifary et al., 2017). In particular, our proposed method is based on Domain Adversarial Nets (DAN), which was originally invented for DA (Ganin et al., 2016) and Xie et al. (2017) demonstrated its efficacy in DG. In addition, Xie et al. (2017) intuitively explained the trade-off between classification accuracy and domain invariance, but they did not suggest any solution to the problem except for carefully tuning a weighting parameter.

Several studies that address DG without utilizing IFL have been conducted. For example, CCSA (Motiian et al., 2017), CIDG (Li et al., 2018b), and CIDDG (Li et al., 2018c) proposed to make use of *semantic alignment*, which attempts to make latent representation given class label ($p(h|y)$) identical within source domains. This approach was originally proposed by Gong et al. (2016) in the DA context, but its efficacy to overcome the trade-off problem is not obvious. CrossGrad (Shankar et al., 2018), which is one of the recent state-of-the-art DG methods, utilizes data augmentation with adversarial examples. However, because the method relies on the assumption that $y$ and $d$ are independent, it might not be directly applicable to our setting.

In DA, Zhang et al. (2013); Gong et al. (2016) address the situation where $p(y)$ changes across the source and target domains by correcting the change of $p(y)$ using unlabeled target data, which is often accomplished at the cost of classification accuracy for the source domain. However, this approach is not applicable (or necessary) to DG because we are agnostic on target domain(s), and this paper is concerned with the change of $p(y)$ within source domains. Instead, we propose to maximize the classification accuracy for source domains while improving the domain invariance.

## 3    PROPOSED METHOD

### 3.1    ACCURACY-CONSTRAINED DOMAIN INVARIANCE

Here we provide the notion of accuracy-constrained domain invariance, which is the maximum domain invariance within a range that does not interfere with the classification accuracy. The reason for the constraint is that the primary purpose of DG is the classification for unseen domains rather than domain itself, and the improvement of the invariance could detrimentally affect the performance.

**Theorem 1** *Let $h = f(x)$, i.e., $h$ is a deterministic mapping of $x$ with a function $f$. We define accuracy-constrained domain invariance as the maximum $H(d|h)$ value under the constraint that*

$H(y|x) = 0$, *i.e., there is no labeling error, and $h$ has as much $y$ information as $x$, i.e., $H(y|h) = H(y|x)$. Accuracy-constrained domain invariance equals $H(d|y)$.*

**Proof 1** *Using the properties of entropy, the following inequation holds:*

$$H(d|h) \leq H(d, y|h) = H(d|y, h) + H(y|h) \leq H(d|h) + H(y|h) \tag{1}$$

*By assumption, $H(y|x) = H(y|h) = 0$, and thus the following inequation holds:*

$$H(d|h) \leq H(d|y) \tag{2}$$

*Thus, the maximum $H(d|h)$ value under the constraints is $H(d|y)$.*

## 3.2 AFLAC

We propose a novel method named AFLAC, which is designed to achieve accuracy-constraind domain invariance. Formally, we denote $f_E(x)$, $q_M(y|h)$, and $q_D(d|h)$ ($E$, $M$, and $D$ are the parameters) as the deterministic encoder, probabilistic model of the label classifier, and that of domain discriminator, respectively. Then, the objective function of AFLAC is described as follows:

$$\min_{E,M} V(E, M) = \mathbb{E}_{x,d,y \sim p(x,d,y)}[\gamma D_{KL}[p(d|y)|q_D(d|h = f_E(x))] - \log q_M(y|h = f_E(x))] \tag{3}$$

$$\min_{D} W(D) = \mathbb{E}_{x,d \sim p(x,d)}[-\log q_D(d|h = f_E(x))] \tag{4}$$

Here $\gamma$ denotes a weighting parameter. Note that, although we cannot obtain true distribution $p(d|y)$, we can use the maximum likelihood estimator of it when $y$ and $d$ are discrete, as is usual with DG.

Here we formally show that AFLAC is intended to achieve $H(d|h) = H(d|y)$ (accuracy-constrained domain invariance) by a Nash equilibrium analysis similar to Goodfellow et al. (2014); Xie et al. (2017). We define $D^*$ and $M^*$ as the solutions to Eq. 3 and Eq. 4 with fixed $E$. They obviously satisfy $q_D^* = p(d|h)$, $q_M^* = p(y|h)$, respectively. Thus, $V$ in Eq. 3 can be written as follows:

$$V(E) = \mathbb{E}[\gamma D_{KL}[p(d|y)|p(d|h)]] + H(y|h) \tag{5}$$

$E^*$, which we define as the solution to Eq. 5 and in Nash equilibrium, satisfies not only $H(y|h) = H(y|x)$ (optimal classification accuracy) but also $\mathbb{E}_{h,y \sim p(h,y)}[D_{KL}[p(d|y)|p(d|h)]] = 0$, which is a sufficient condition for $H(d|h) = H(d|y)$ by the definition of the conditional entropy.

## 4 EXPERIMENTS

### 4.1 DATASETS

**BMNISTR** We created the Biased Rotated MNIST dataset (BMNISTR) by modifying the sample size of the popular benchmark dataset MNISTR (Ghifary et al., 2015), such that the class distribution differed among the domains. In MNISTR, each domain was created by rotating images by 15 degree increments: 0, 15, ..., 75 (referred to as M0, ..., M75). We created four variants of MNISTR that have different types of domain-class dependency, referred to as BMNISTR-1 through BMNISTR-3. As shown in Table 1-left, BMNISTR-1, -2 have similar trends but different degrees of dependency, whereas BMNISTR-1 and BMNISTR-3 differ in terms of their trends. In training, we employed a leave-one-domain-out setting (Ghifary et al., 2015): we trained the models on five of the six domains and tested them using the remaining one.

**WISDM** WISDM contains sensor data of accelerometers of six human activities (walking, jogging, upstairs, downstairs, sitting, and standing) performed by 36 users (domains). WISDM has the dependency for the reason noted in Sec. 1. we randomly selected <10 / 26> and <26 / 10> users as <source / target> domains, and split the source data into training and validation data.

### 4.2 BASELINES

We compared AFLAC with the following methods. **(1) CNN** is a vanilla convolutional networks trained on the aggregation of data from all source domains. **(2) DAN (Xie et al., 2017)** is expected

Table 1: Left: Sample sizes for each domain-class pair in BMNISTR. Those for the classes 0∼4 are variable across domains, whereas the classes 5∼9 have identical sample sizes across domains. Right: Mean F-measures for the classes 0∼4 and classes 5∼9 with the target domain M0. RI denotes relative improvement of AFLAC to AFLAC-Abl

| Dataset | Class | M0 | M15 | M30 | M45 | M60 | M75 |
|---|---|---|---|---|---|---|---|
| BMNISTR-1 | 0∼4 | 100 | 85 | 70 | 55 | 40 | 25 |
| | 5∼9 | 100 | 100 | 100 | 100 | 100 | 100 |
| BMNISTR-2 | 0∼4 | 100 | 90 | 80 | 70 | 60 | 50 |
| | 5∼9 | 100 | 100 | 100 | 100 | 100 | 100 |
| BMNISTR-3 | 0∼4 | 100 | 25 | 100 | 25 | 100 | 25 |
| | 5∼9 | 100 | 100 | 100 | 100 | 100 | 100 |

| Dataset | Class | CNN | DAN | CIDDG | AFLAC-Abl | AFLAC | RI |
|---|---|---|---|---|---|---|---|
| BMNISTR-1 | 0∼4 | 83.86 | 84.54 | 87.50 | 87.46 | **90.62** | 3.6% |
| | 5∼9 | 83.90 | 85.24 | 87.46 | 86.46 | **88.10** | 1.9% |
| BMNISTR-2 | 0∼4 | 82.54 | 85.30 | 87.64 | 88.60 | **89.64** | 1.2% |
| | 5∼9 | 82.18 | 85.80 | 86.74 | 87.60 | **89.04** | 1.6% |
| BMNISTR-3 | 0∼4 | 71.26 | 79.22 | 76.76 | 76.56 | **80.02** | 4.5% |
| | 5∼9 | 78.62 | **83.14** | 82.64 | 82.94 | 82.80 | -0.2% |

(a) BMNISTR-1, M0    (b) BMNISTR-1, M75    (c) WISDM, 10 users    (d) WISDM, 26 users

Figure 2: Classification Accuracy with various $\gamma$. Each caption shows dataset and target name.

to generalize across domains utilizing domain-invariant representation, but it can be affected by the trade-off as pointed out by Xie et al. (2017). **(3) CIDDG** is our re-implementation of Li et al. (2018c), which is designed to achieve semantic alignment on adversarial training. Additionally, we used **(4) AFLAC-Abl**, which is a version of AFLAC modified for ablation studies. AFLAC-Abl replaces $D_{KL}[p(d|y)|q_D(d|h)]$ in Eq. 3 of $D_{KL}[p(d)|q_D(d|h)]$, thus it attempts to learn the representation that is completely invariant to domains or make $H(d|h) = H(d)$ hold as well as DAN. Comparing AFLAC and AFLAC-Abl, we measured the genuine effect of taking domain-class dependency into account. When training AFLAC and AFLAC-Abl, we cannot obtain true $p(d|y)$ and $p(d)$, hence we used their maximum likelihood estimators for calculating the KLD terms.

## 4.3 RESULTS

We first investigated the extent to which domain-class dependency affects the performance of domain-invariance-based methods. In Table 1-right, we compared the mean F-measures for the classes 0 through 4 and classes 5 through 9 in BMNISTR with the target domain M0. Recall that the sample sizes for the classes 0∼4 are variable across domains, whereas the classes 5∼9 have identical sample sizes across domains. The F-measures show that AFLAC outperformed baselines in most dataset-class pairs, which supports that the dependency reduces the performance of IFL methods and that AFLAC can mitigate the problem. Further, the relative improvement of AFLAC to AFLAC-Abl is more significant for the classes 0∼4 than for 5∼9 in BMNISTR-1 and BMNISTR-3, suggesting that AFLAC tends to increase performance more significantly for classes in which the dependency occurs. Moreover, the improvement is more significant in BMNISTR-1 than in BMNISTR-2, suggesting that the stronger the dependency is, the lower the performance of domain-invariance-based methods becomes. Finally, although the dependencies of BMNISTR-1 and BMNISTR-3 have different trends, AFLAC improved the F-measures in both datasets.

Next we investigated the relationship between the strength of regularization and performance. Figures 2-(b, c, d) show that the accuracy gaps of AFLAC-Abl and AFLAC increase with strong regularization (such as when $\gamma = 10$), suggesting that AFLAC, as it was designed, does not tend to reduce accuracy with strong regularizer, and thus AFLAC is robust toward hyperparameter choice.

## 5 CONCLUSION

In this paper, we proposed a novel method AFLAC, which maximizes domain invariance within a range that does not interfere with classification accuracy on adversarial training. Empirical validations show the superior DG performance of AFLAC to the baseline methods, supporting the importance of the domain-class dependency in domain generalization tasks and the efficacy of the proposed method for overcoming the issue.

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
