# OpenReview forum: "Adversarial Feature Learning under Accuracy Constraint for Domain Generalization"
_ICLR.cc/2019/Workshop/LLD — LLD 2019_

### Official Review · AnonReviewer2 · 2019-04-07
**Well written and straight to the point paper**

**Rating:** 4
**Confidence:** 2

**Review:**

In this paper, the authors address the problem of potential dependency between classes and domains in domain generalization. Based on remarks from Xie et al. (2017), they propose a new objective function AFLAC, which (provably) enforces accuracy-constrained domain invariance at its optimum.

The positioning towards related works is clear, the technical analysis is brief but precise. The experiments are well conducted and show both an improvement over the state of the art (including recent approaches) and robustness toward the choice of the hyperparameter.

A longer version of this work could include a more detailed scenario of the practical application of this method, as well as experiments on more real datasets (unlike BMNISTR) illustrating the precise advantage of AFLAC on this data.

---

### Official Review · AnonReviewer1 · 2019-04-10
**A good paper proposing an objective function for domain invariant representation learning under accuracy constraints**

**Rating:** 4
**Confidence:** 2

**Review:**

The paper provides an analysis on the tradeoff between accuracy and domain invariant representation learning, and accordingly proposes an objective function for domain invariant representation learning under accuracy constraints. The paper is well-written and novel.

Pros:
- The methods are well-grounded mathematically.
- Well-selected baselines and ablation studies. It is good to see that the authors included an ablation study on the term D_KL[p(d|y)|qD(d|h)], and verified the effectiveness of the domain-class dependence assumption.

Cons:
- The datasets are relatively small.

Others:
Please check the Equation 1 in proof 1. I think the final term should have been H(d|y) + H(y|h). It seems like a typo.

---

### Decision · Program_Chairs · 2019-04-16
**Acceptance Decision**

Accept